# Emergent Plants Improve Nitrogen Uptake Rates by Regulating the Activity of Nitrogen Assimilation Enzymes

**DOI:** 10.3390/plants14101484

**Published:** 2025-05-15

**Authors:** Yu Hong, Ruliang Liu, Wenhua Xiang, Pifeng Lei, Xi Fang

**Affiliations:** 1College of Ecology and Environment, Central South University of Forestry and Technology, Changsha 410004, China; hongyu_2007@163.com (Y.H.); xiangwh2005@163.com (W.X.);; 2Institute of Agricultural Resources and Environment, Ningxia Academy of Agriculture and Forestry Science, Yinchuan 750002, China; ruliang_liu@126.com; 3Huitong National Field Station for Scientific Observation and Research of Chinese Fir Plantation Ecosystem in Hunan Province, Huaihua 438107, China

**Keywords:** nitrogen uptake rate, NH_4_^+^/NO_3_^−^ ratio, nitrogen assimilation enzyme activity, *Phragmites australis*, *Typha orientalis*

## Abstract

Effectively utilizing aquatic plants to absorb nitrogen from water bodies and convert it into organic nitrogen via nitrogen assimilation enzyme activity reduces water nitrogen concentrations. This serves as a critical strategy for mitigating agricultural non-point source pollution in the Yellow River Basin However, emergent plants’ rate and mechanism of uptake of different forms of nitrogen remain unclear. This study determined the nitrogen uptake rates, nitrogen assimilation activities, root properties, and photosynthetic parameters of four emergent plants, *Phragmites australis*, *Typha orientalis*, *Scirpus validus*, and *Lythrum salicaria*, under five NH_4_^+^/NO_3_^−^ ratios (9:1, 7:3, 5:5, 3:7, and 1:9) using ^15^N hydroponic simulations. The results demonstrated that both the form of nitrogen and the plant species significantly influenced the nitrogen uptake rates of emergent plants. In water bodies with varying NH_4_^+^/NO_3_^−^ ratios, *P. australis* and *T. orientalis* exhibited significantly higher inorganic nitrogen uptake rates than *S. validus* and *L. salicaria*, increasing by 11.83–114.69% and 14.07–130.46%, respectively. When the ratio of NH_4_^+^/NO_3_^−^ in the water body was 9:1, the uptake rate of inorganic nitrogen by *P. australis* reached its peak, which was 729.20 μg·N·g^−1^·h^−1^ DW (Dry Weight). When the ratio of NH_4_^+^/NO_3_^−^ was 5:5, the uptake rate of *T. orientalis* was the highest, reaching 763.71 μg·N·g^−1^·h^−1^ DW. The plants’ preferences for different forms of nitrogen exhibited significant environmental plasticity. At an NH_4_^+^/NO_3_^−^ ratio of 5:5, *P. australis* and *T. orientalis* preferred NO_3_^−^-N, whereas *S. validus* and *L. salicaria* favored NH_4_^+^-N. The uptake rate of NH_4_^+^-N by the four plants was significantly positively correlated with glutamine synthetase and glutamate synthase activities, while the uptake rate of NO_3_^−^-N was significantly positively correlated with NR activity. These findings indicate that the nitrogen uptake and assimilation processes of these four plant species involve synergistic mechanisms of environmental adaptation and physiological regulation, enabling more effective utilization of different nitrogen forms in water. Additionally, the uptake rate of NH_4_^+^-N by *P. australis* and *T. orientalis* was significantly positively correlated with glutamate dehydrogenase (GDH), suggesting that they are better adapted to eutrophication via the GDH pathway. The specific root surface area plays a crucial role in regulating the nitrogen uptake rates of plants. The amount of nitrogen uptake exerted the greatest total impact on the nitrogen uptake rate, followed by root traits and nitrogen assimilation enzymes. Therefore, there were significant interspecific differences in the uptake rates of and physiological response mechanisms of emergent plants to various nitrogen forms. It is recommended to prioritize the use of highly adaptable emergent plants such as *P. australis* and *T. orientalis* in the Yellow River irrigation area.

## 1. Introduction

Nitrogen (N) plays a crucial role in plants’ physiological metabolism, growth, and development, as well as in the nutrient cycling of ecosystems [1]. As a key component of aquatic ecosystems, aquatic plants are capable of efficiently absorbing N from water and assimilating it into organic N, such as amino acids and proteins, via N assimilation enzyme activity. This process not only supports plants’ growth and metabolism, but also contributes to reducing N concentrations in the water [2]. Consequently, aquatic plants are essential for N cycling and water quality improvement in aquatic systems, making them a significant current focus of research [3]. Different forms of N significantly influence the efficiency of N metabolism and the ecological functions of plants by modulating the activities of key enzymes that are involved in N assimilation [4]. Plants convert inorganic N into organic N via enzyme systems, including nitrate reductase (NR), nitrite reductase (NiR), glutamine synthetase (GS), glutamate synthase (GOGAT), and glutamate dehydrogenase (GDH). This process is fundamental to N uptake and assimilation in plants [5]. As illustrated in Figure 1, the N uptake and assimilation processes in plants proceed as follows: Aquatic plants are proficient at directly absorbing dissolved nitrogenous compounds from the substrate through their roots, especially ammonium nitrogen (NH_4_^+^-N) and nitrate nitrogen (NO_3_^−^-N). The absorbed N is subsequently transported to various plant organs, such as stems and leaves. Through enzymatic activities associated with N assimilation, these inorganic N forms are converted into organic molecules, such as amino acids and proteins [6]. This process not only facilitates nutrient storage within the plant, but also significantly contributes to its growth and metabolism, thereby reducing nitrogenous pollutants in aquatic environments [2,7]. The primary biochemical pathways that are involved include the following: (1) Nitrate reduction, which occurs when NR in the cytoplasm reduces nitrate ions (NO_3_^−^) to nitrite ions (NO_2_^−^), which are further reduced to ammonium ions (NH_4_^+^) by NiR located in the chloroplasts and plastids of root cells. (2) Ammonium assimilation, a process in which GS in the cytoplasm and plastids catalyzes the reaction between NH_4_^+^ and glutamic acid (Glu) to form glutamine (Gln). Subsequently, GOGAT in the chloroplasts and root cells catalyzes the reaction between Gln and α-ketoglutaric acid (α-KG) to produce two molecules of Glu. (3) The regulation of ammonium availability, which occurs under a high ammonium concentration, during which GDH in the mitochondria and cytoplasm facilitates the binding of α-KG with NH_4_^+^ to generate Glu. Conversely, during ammonium starvation, GDH can catalyze the reverse reaction, decomposing Glu into α-KG and NH_4_^+^ [8,9].

The activities of the key enzymes that are involved in N assimilation in aquatic plants are influenced by both the N forms in the water and environmental factors. NO_3_^−^-N predominantly induces NR and NiR activities [10], while NH_4_^+^-N predominantly induces GS, GOGAT, and GDH activities [11,12]. NR serves as a rate-limiting enzyme for the assimilation of NO_3_^−^-N, catalyzing the reduction of NO_3_^−^ to NO_2_^−^. NiR is a key enzyme that reduces NO_2_^−^ to NH_4_^+^, and its activity is positively correlated with NR activity [13]. Under NO_3_^−^-N supply conditions, the NR activity is relatively high. However, in high NH_4_^+^ environments, the activities of both NR and NiR may be inhibited [14]. Together, GS and GOGAT constitute the GS/GOGAT cycle, which is a critical metabolic pathway for plants to assimilate NH_4_^+^ into organic N [15]. Due to the synergistic effects within this cycle, GOGAT activity is usually positively correlated with GS activity, and both GS and GOGAT activities are generally higher in plants when NH_4_^+^-N is the primary N source [16]. The GDH pathway is one of the key pathways involved in N and energy metabolism. GDH activity may significantly increase under stress conditions, serving as an important mechanism for ammonium detoxification in plants [12].

The migration, transformation, and bioavailability of different N forms (NH_4_^+^-N, NO_3_^−^-N, etc.) in aquatic environments vary, potentially driving aquatic plant roots to adopt distinct strategies for N uptake and assimilation to sustain normal growth and development [17]. These pathways and strategies are influenced by a combination of factors, such as the plant species, N morphology in the environment, and ammonium-to-nitrate ratio (NH_4_^+^/NO_3_^−^) [18]. Different plant species demonstrate distinct nutrient preferences and utilization strategies across varying habitats [17]. Research has indicated that the absorption of amino acids and utilization of N in plants are influenced by the type of N source and exhibit a certain degree of adaptability to environmental changes in N forms [19]. In sandy constructed wetlands, varying NH_4_^+^/NO_3_^−^ ratios directly influence the plant diversity, N removal efficiency, and ecosystem stability [20]. Increasing the proportion of NH_4_^+^ can enhance the GOGAT and GS activities in plants that are involved in NH_4_^+^ assimilation, thereby accelerating N metabolism [11].

Different N forms exert significant regulatory effects on the morphology and function of roots. NH_4_^+^ generally enhances the accumulation of root biomass, leading to a significant increase in the total root length, root surface area, and lateral root density [21]. The strategy of sacrificing the taproot and enhancing lateral root development may be linked to the higher energy demands for NO_3_^−^ assimilation [22], which compels plants to enhance their N capture efficiency by increasing their root absorption area. Modifying the specific root length (SRL) and specific root surface area (SRA) to adjust their nutrient uptake capacity constitutes a key adaptive mechanism for plants in response to environmental changes [23]. In addition, N is a crucial constituent of chlorophyll and photosynthetic enzymes, and varying N forms significantly impact plants’ photosynthetic efficiency [24].

Although the uptake rates and physiological characteristics of different N forms in aquatic plants have been studied, there is still a paucity of reports regarding the regulatory mechanisms of N metabolism under mixed-N (NH_4_^+^-N and NO_3_^−^-N) conditions. Specifically, the coupling mechanism between the activities of the key enzymes that are involved in N assimilation, root traits, photosynthetic characteristics, and N uptake rates remains to be fully elucidated. Consequently, it is crucial to investigate the uptake rates and physiological synergistic mechanisms of NH_4_^+^-N and NO_3_^−^-N uptake by aquatic plants in aquatic environments. The Ningxia Yellow River irrigation area serves as a crucial irrigation zone and commercial grain production base in northwest China, contributing over two-thirds of the output value of grain crops and agriculture in Ningxia. It also plays a significant role in driving regional economic development [25]. Long-term extensive agricultural management practices, such as the overuse of chemical fertilizers and excessive irrigation, have led the water bodies in the Yellow River Basin to face an increasingly severe issue of agricultural non-point source pollution [26]. Aquatic phytoremediation technology for the removal of excess N from agricultural water bodies exhibits a remarkable purification efficiency and resource recycling potential [27]. This constitutes a pivotal strategy for controlling agricultural non-point source pollution in the Yellow River irrigation area and preventing and alleviating potential pollution risks to the Yellow River [28]. Furthermore, it signifies an emerging priority in the advancement of ecological and environmental research and pollution prevention and control technologies within the basin [29]. Emergent plants are a category of aquatic plants whose roots are embedded in mud, with the lower part of the stem or base being submerged in water and photosynthetic structures, such as stems and leaves, protruding above the water surface [30]. These plants display a wide ecological range and fulfill multiple ecological functions, including wind resistance, pollution tolerance, decontamination capacity, and the alleviation of eutrophication in aquatic ecosystems. Furthermore, they exhibit superior resilience compared with floating and submerged plants [31] and are widely distributed across the wetlands in northwest China’s arid region. The synergistic effect of pollutant removal that can be achieved through the absorption and assimilation capabilities of emergent plants, in conjunction with the degradation and transformation activities of rhizosphere microorganisms [32], constitutes an environmentally friendly technology that is characterized by low resource input, minimal energy consumption, and the prevention of secondary pollution. This approach has emerged as a pivotal strategy for ensuring sustained water quality in irrigation regions.

Therefore, in this study, four emergent plants, *P. australis*, *T. orientalis*, *S. validus*, and *L. salicaria*, were used as research objects. In the experimental water, which was maintained at a constant N concentration, five NH_4_^+^/NO_3_^−^ ratios (9:1, 7:3, 5:5, 3:7, and 1:9) labeled with ^15^N isotopes were established. The N uptake rate, N assimilation enzyme activity, root properties, and photosynthetic parameters of the plants were measured to verify the following two hypotheses: (1) different forms of N significantly affect the N uptake rate and N-assimilation-related enzyme activities of emergent plants; and (2) the main factors influencing the different N uptake rates of emergent plants are the plant’s root attributes, N assimilation enzyme activity, and photosynthetic parameters. This paper analyzes the responses of the N uptake rate and physiological characteristics of emergent plants to different forms of N (NH_4_^+^-N, NO_3_^−^-N); clarifies the adaptation strategies of plants to N forms; identifies the removal mechanism of N pollution by emergent plants; provides a scientific basis for optimizing the efficient purification of emergent plants and using them to remediate N pollution in water; and deepens our understanding of the correlation mechanism of “nitrogen-form–plant-species–nitrogen-uptake rates” in the process of N removal from water.

## 2. Results

### 2.1. Nitrogen Uptake Rates of Emergent Plants in Water with Different NH_4_^+^/NO_3_^−^ Ratios

The N form, plant species, NH_4_^+^/NO_3_^−^ ratio, and their interactions significantly affected the N uptake rates of the four emergent plants (*p* < 0.001). Among these factors, the N form was the most influential, followed by the plant species (Table 1).

As illustrated in Figure 2, in the experimental water with different NH_4_^+^/NO_3_^−^ ratios, the uptake rates of inorganic N of *P. australis* and *T. orientalis* were significantly higher than those of the other two plant species (*p* < 0.05). When the NH_4_^+^/NO_3_^−^ ratio was 9:1, the uptake rate of inorganic N peaked at 729.20 μg N·g^−1^·h^−1^ DW for *P. australis* and 477.36 μg N·g^−1^·h^−1^ DW for *S. validus*. In contrast, when the NH_4_^+^/NO_3_^−^ ratio was 5:5, the highest uptake rates of inorganic N were observed in *T. orientalis* (763.71 μg N·g^−1^·h^−1^ DW) and *L. salicaria* (609.92 μg N·g^−1^·h^−1^ DW).

When NH_4_^+^-N was dominant (NH_4_^+^/NO_3_^−^ ratios of 9:1 and 7:3), *P. australis*, *T. orientalis*, *S. validus*, and *L. salicaria* exhibited a preference for absorbing NH_4_^+^-N, and the ratios of NH_4_^+^-N to NO_3_^−^-N uptake rates were all greater than 1. When the NH_4_^+^/NO_3_^−^ ratio was 1:9, the four emergent plants preferred to absorb NO_3_^−^-N, and the ratios of NH_4_^+^-N to NO_3_^−^-N uptake rates were all less than 1. This indicates that the inorganic N uptake rates of these four species exhibit a certain degree of environmental plasticity. When the NH_4_^+^/NO_3_^−^ ratio was 5:5, the four plants showed varying preferences for different N forms. *P. australis* and *T. orientalis* preferred NO_3_^−^-N absorption, while *S. validus* and *L. salicaria* favored NH_4_^+^-N absorption.

### 2.2. Nitrogen Assimilation Enzyme Activities of Emergent Plants in Water with Different NH_4_^+^/NO_3_^−^ Ratios

As illustrated in Figure 3, with increasing NH_4_^+^/NO_3_^−^ ratios, the contents of soluble protein (SP) and activity of NR in the four emergent plants decreased. In contrast, the GS activity increased, while the GDH activity initially increased and then decreased. There was no obvious regularity in the changes in NiR and GOGAT activities. When the NH_4_^+^/NO_3_⁻ ratio was 1:9, the SP contents in *T. orientalis*’s stems, leaves, and roots reached their highest levels, at 16.96 mg·g^−1^ FW (Fresh Weight) in the stems and leaves and 14.83 mg·g^−1^ FW in the roots. Similarly, the NR activities in *T. orientalis*’s stems, leaves, and roots were also the highest, at 43.08 μmol·h^−1^·g^−1^ FW in the stems and leaves and 46.62 μmol·h^−1^·g^−1^ FW in the roots. At an NH_4_^+^/NO_3_⁻ ratio of 9:1, the GS activities in *P. australis*’s stems and leaves, as well as *T. orientalis*’s roots, were the highest, reaching 39.52 μmol·h^−1^·g^−1^ FW in the stems and leaves and 27.34 μmol·h^−1^·g^−1^ FW in the roots. When the NH_4_^+^/NO_3_⁻ ratio was 7:3, the GDH activities in *S. validus*’s stems and leaves and *L. salicaria*’s roots were the highest, at 66.15 μmol·h^−1^·g^−1^ FW in the stems and leaves and 78.62 μmol·h^−1^·g^−1^ FW in the roots. Furthermore, when the NH_4_^+^/NO_3_⁻ ratio was 5:5, the NiR activity in *T. orientalis*’s stems and leaves was relatively high, at 44.53 μmol·h^−1^·g^−1^ FW. At an NH_4_^+^/NO_3_⁻ ratio of 7:3, the NiR activity in *T. orientalis*’s roots was higher, at 36.83 μmol·h^−1^·g^−1^ FW. Lastly, when the NH_4_^+^/NO_3_⁻ ratio was 7:3, the GOGAT activities in *P. australis*’s stems and leaves, as well as *T. orientalis*’s roots, were the highest, at 37.80 μmol·h^−1^·g^−1^ FW in the stems and leaves and 22.80 μmol·h^−1^·g^−1^ FW in the roots.

At varying NH_4_^+^/NO_3_^−^ ratios, the contents of SP in the stems and leaves of the four emergent plants, as well as the activities of NiR, GS, and GOGAT, were all higher than those in the roots. Conversely, the GDH activities in the stems and leaves were lower than those in the roots for all four species. The NR activity in the stems and leaves of *P. australis* exceeded that in the roots, whereas in *T. orientalis*, the NR activity was lower in the stems and leaves than in the roots. The NR activities of *S. validus* and *L. salicaria* were sometimes higher in the stems and leaves and sometimes in the roots (Figure 3).

### 2.3. Influencing Factors on Nitrogen Uptake Rates of Different Emergent Plants

As shown in Figure 4a, the NH_4_^+^-N uptake rate of *P. australis* exhibited a significantly positive correlation with the NH_4_^+^/NO_3_^−^ ratio, NH_4_^+^-N uptake amount, GS activity, GOGAT activity, and GDH activity (*p* < 0.05), while it was significantly negatively correlated with the NO_3_^−^-N uptake amount, SP content, NR activity, and biomass (*p* < 0.05). The NO_3_^−^-N uptake rate demonstrated a significantly positive correlation with the NO_3_^−^-N uptake amount, SP content, and NR activity (*p* < 0.01), while it was significantly negatively correlated with the NH_4_^+^-N uptake amount, NH_4_^+^/NO_3_^−^ ratio, GS activity, GOGAT activity, and GDH activity (*p* < 0.05). Additionally, the inorganic N uptake rate was significantly positively correlated with the specific root surface area (SRA) and total nitrogen (TN) content (*p* < 0.05), whereas it was negatively correlated with the biomass and root tissue density (RTD) (*p* < 0.05).

As shown in Figure 4b, the NH_4_^+^-N uptake rate of *T. orientalis* exhibited a significantly positive correlation with the NH_4_^+^/NO_3_^−^ ratio, NH_4_^+^-N uptake amount, GS activity, GOGAT activity, and GDH activity (*p* < 0.05). Conversely, it was significantly negatively correlated with the NO_3_^−^-N uptake amount, SP content, and NR activity (*p* < 0.01). The NO_3_^−^-N uptake rate demonstrated a significant positive correlation with the NO_3_^−^-N uptake amount, SP content, and NR activity (*p* < 0.01), while it was significantly negatively correlated with the NH_4_^+^-N uptake amount, NH_4_^+^/NO_3_^−^ ratio, GS activity, GOGAT activity, and GDH activity (*p* < 0.05). Additionally, the uptake rate of inorganic N was significantly positively correlated with the SRA and specific root length (SRL) (*p* < 0.05), whereas it was negatively correlated with the RTD, biomass, and root diameter (AD) (*p* < 0.05).

As illustrated in Figure 4c, the NH_4_^+^-N uptake rate of *S. validus* showed a significantly positive correlation with the NH_4_^+^/NO_3_^−^ ratio, GS activity, GOGAT activity, GDH activity, NH_4_^+^-N uptake amount, SRL, and SRA (*p* < 0.05). In contrast, it exhibited a significantly negative correlation with the NO_3_^−^-N uptake amount, SP, NR, RTD, and AD (*p* < 0.05). Furthermore, the NO_3_^−^-N uptake rate was significantly positively correlated with the NO_3_^−^-N uptake amount and NR activity (*p* < 0.05), whereas it was significantly negatively correlated with the GS activity, GOGAT activity, and GDH activity (*p* < 0.05). Lastly, the inorganic N uptake rate demonstrated a significantly positive correlation with the NH_4_^+^/NO_3_^−^ ratio, GS activity, SRA, and SRL (*p* < 0.05), while it was significantly negatively correlated with the SP content, RTD, NR activity, NiR activity, and AD (*p* < 0.05).

As illustrated in Figure 4d, the NH_4_^+^-N uptake rate of *L. salicaria* showed a significantly positive correlation with the NH_4_^+^/NO_3_^−^ ratio, NH_4_^+^-N uptake amount, GS activity, and GOGAT activity (*p* < 0.05), whereas it exhibited a significantly negative correlation with the NO_3_^−^-N uptake amount, SP content, and NR activity (*p* < 0.01). The NO_3_^−^-N uptake rate was significantly positively correlated with the NO_3_^−^-N uptake amount, SP content, and NR activity (*p* < 0.01), while it was significantly negatively correlated with the NH_4_^+^/NO_3_^−^ ratio, NH_4_^+^-N uptake amount, GS activity, and GOGAT activity (*p* < 0.01). Furthermore, the inorganic N uptake rate demonstrated a significantly positive correlation with the NH_4_^+^-N uptake amount and GS activity (*p* < 0.05), but it was significantly negatively correlated with the SP content and NR activity (*p* < 0.05).

### 2.4. The Main Driving Factors of the Nitrogen Uptake Rates of Emergent Plants

The results of our redundancy analysis (RDA) (Figure 5) revealed that the interpretation rates of the first and second axes were 57.81% and 34.30%, respectively, with a cumulative interpretation rate of 92.11%. The main indicators influencing the N uptake rates of emergent plants, ranked from most to least significant, were as follows: the NO_3_^−^-N uptake amount in stems and leaves, TN content in stems and leaves, NH_4_^+^-N uptake amount in stems and leaves, SRA, NH_4_^+^-N uptake amount in roots, NR activity in stems and leaves, NiR activity in roots, SP content in stems and leaves, SP content in roots, and NR activity in roots.

Among these factors, the NH_4_^+^-N uptake rate exhibited a significantly positive correlation with the TN content in stems and leaves, NH_4_^+^-N uptake amount, and SRA (*p* < 0.05). The NO_3_^−^-N uptake rate demonstrated a significantly positive correlation with the NO_3_^−^-N uptake amount in stems and leaves, TN content in stems and leaves, SRA, NR activity in stems and leaves, NiR activity in roots, and SP content (*p* < 0.05). Furthermore, the inorganic N uptake rate was significantly positively correlated with the NO_3_^−^-N uptake amount in stems and leaves, TN content in stems and leaves, NH_4_^+^-N uptake amount, SRA, NR activity, NiR activity in roots, and SP content (*p* < 0.05).

The partial least squares path model (PLS-PM) was utilized to investigate the potential direct and indirect effects of the N uptake amount, root traits, N assimilation enzymes, and photosynthetic parameters on the N uptake rates of emergent plants (Figure 6). The PLS-PM analysis results demonstrated that the N uptake amount and root traits had a direct and significantly positive influence on the N uptake rate (*p* < 0.001), with path coefficients of 0.786 and 0.460, respectively. The plants’ TN contents contributed more than 0.8 to the N uptake amount, while the SRL and SRA accounted for over 0.8 of the influence on root traits. Furthermore, the N assimilation enzymes had a direct and significant negative effect on the N uptake rate (*p* < 0.01), with a path coefficient of −0.194. The effects of NiR, SP, and NR on the N assimilation enzymes were all greater than 0.8. In addition, the root traits showed a direct and significant positive effect on the N assimilation enzymes, while the N assimilation enzymes also had a direct and significant positive effect on the N uptake (*p* < 0.001). The main effects analysis of the PLS-PM indicated that the N uptake amount exerted the greatest total impact on the N uptake rate, followed by root traits and N assimilation enzymes (Figure 6).

## 3. Discussion

### 3.1. Effects of Different Forms of Nitrogen on Nitrogen Uptake Rate of Emergent Plants

Plant growth is optimized under a balanced supply of NH_4_^+^-N and NO_3_^−^-N, and the efficiency of N acquisition is determined by the ionic availability in the habitat [17,33]. In this study, we have demonstrated that both the form of N and the plant species significantly affected the N uptake rate of four emergent plants in the Yellow River irrigation area (*p* < 0.001) (Table 1). When NH_4_^+^-N was predominant, the plants showed a clear preference for NH_4_^+^-N absorption. When NO_3_^−^-N was predominant, the plants favored the uptake of NO_3_^−^-N (Figure 2). These results suggest that different N forms significantly affected the N uptake rates of these emergent plants, and the N uptake rates exhibited a certain degree of environmental adaptability. These findings align with those reported by Daryanto et al. (2019) [34], thus confirming part of our first hypothesis. This phenomenon can be explained based on the following three aspects:

(1)Plants require different levels of energy for the absorption and assimilation of various N forms, and the associated metabolic pathways differ accordingly. NH_4_^+^ is directly involved in amino acid synthesis and requires less energy, thus eliminating the additional energy expenditure that is associated with NO_3_^−^ reduction [22]. While the absorption and assimilation of NO_3_^−^ demand more energy, its low toxicity reduces the metabolic burden on plants [35]. Plants primarily rely on ammonium transporters for NH_4_^+^ uptake, whereas nitrate transporters are essential for NO_3_^−^ uptake. Consequently, the predominant N form in the water may enhance the uptake of that specific N form by modulating the expression levels of the corresponding transporters [36].(2)Changes in environmental factors directly or indirectly influence plants’ physiological activities, thereby modulating their N uptake rates. For example, high temperatures significantly reduce the relative growth rate, N uptake rate, NR activity, and photosynthetic parameters of aquatic plants [37]. Factors such as varying light intensities, nutrient concentrations, and hydraulic loads also influence the absorption efficiency of plants [38].(3)To adapt to varying environmental conditions, aquatic plants modulate their N uptake, assimilation, and transformation via their growth traits (e.g., the root structure and photosynthetic capacity) and interactions with microorganisms. This not only meets their growth demands, but also significantly decreases the N concentrations in water bodies, thereby reflecting a certain degree of environmental adaptability [39,40]. Studies have shown that aquatic plants can increase their direct N uptake and induce changes in N cycling and their microenvironment by altering their root morphology, growth characteristics, leaf biomass, and rhizosphere conditions, thus reducing N pollution in riparian waters [41]. The N assimilation efficiency of aquatic plants is closely linked to their root-to-shoot ratio. Aquatic plants indirectly affect N removal by microorganisms through modifying tailwater quality parameters and can also mediate N transformation by modulating bacterial community structures [42]. In addition, as the N load in sewage increases, plants’ N preference shifts from NH_4_^+^-N to NO_3_^−^-N. The elevated NR activity that is detected in downstream river plants provides evidence for their enhanced nitrate assimilation capacity and preference [17].

### 3.2. Effects of Different Forms of Nitrogen on Nitrogen Assimilation Enzyme Activity in Emergent Plants

Different forms of N substantially influence the activities of N-assimilation-related enzymes in plants. Through these complex adaptive adjustments, plants sustain their growth and development under varying N supply conditions [14]. In this study, different forms of N significantly affected the activity of N-assimilation-related enzymes in plants (Figure 3), which is largely in agreement with the findings reported by Chen et al. (2024) [43] and supports the second part of our first hypothesis. We found that, as the NH_4_^+^/NO_3_^−^ ratio in the water increased, the NR and SP contents in the emergent plants decreased, whereas the GS activity increased. Moreover, the GDH activity exhibited an initial increase, followed by a decrease. This phenomenon may be explained by means of three potential factors, as follows:(1)NR is a key enzyme involved in the assimilation of NO_3_^−^ by plants, and its activity is influenced by the form of N. Under varying N supply conditions, plants from different ecological groups exhibit significantly different NR activities [44]. When NH_4_^+^-N is supplied, the NR activity in plant roots remains low. In contrast, when NO_3_^−^-N is supplied, the NR activity in plant roots is significantly higher. This may be attributed to NH_4_^+^ competing with NO_3_^−^ for absorption sites, thereby influencing the uptake of NO_3_^−^ and the associated NR activity of plants [45]. In addition, NO_3_^−^ can be assimilated in the roots and subsequently transported to the aerial parts of the plant for further assimilation. This process facilitates the rational and efficient utilization of the carbon skeleton across the entire system. Notably, NR acts as a rate-limiting enzyme, and its activity plays a critical role in regulating N metabolism and protein synthesis [46]. In this study, an increase in the NH_4_^+^/NO_3_^−^ ratio was found to be associated with decreased NR activities and reduced SP contents in the four emergent plants, thereby providing further support for the findings of this study.(2)GS plays a crucial role in plants’ N metabolism, primarily by mediating the assimilation of NH_4_^+^ [47]. Elevated NH_4_^+^ levels stimulate GS activity, leading to more efficient conversion of NH_4_^+^ into organic N compounds [11]. Studies have demonstrated that plants exhibit significant growth advantages and competitiveness under conditions with a high NH_4_^+^/NO_3_^−^ ratio. This can be attributed to their efficient uptake of NH_4_^+^ and the increased activity of enzymes such as GS [4]. In addition, the genes that are most significantly influenced by NO_3_^−^ and NH_4_^+^ treatment were those that are involved in glutamine metabolism, further supporting the link between changes in GS activity and N assimilation in plants [48]. In this study, an increase in the NH_4_^+^/NO_3_^−^ ratio was correlated with enhanced GS activities in the four emergent plants.(3)GDH plays a crucial role in NH_4_^+^ assimilation and glutamate synthesis in plants, thereby modulating the balance of N metabolism and, in turn, significantly contributing to the plants’ growth, development, and stress adaptation [12]. As the NH_4_^+^/NO_3_^−^ ratio increases, plants may mitigate the adverse effects of NH_4_^+^ by enhancing their GDH activity; however, when this ratio becomes excessively high, the GDH activity may decline due to impaired energy supply and a disrupted metabolic balance [11].

### 3.3. Factors Influencing Nitrogen Uptake Rate of Different Emergent Plants

The N uptake rates of different plants are closely linked to their root traits and the activity of N assimilation enzymes. These factors interact synergistically, thus affecting the efficiency of N uptake and utilization in plants [49]. In this study, the NH_4_^+^-N uptake rates of the four emergent plants were significantly positively correlated with the NH_4_^+^/NO_3_^−^ ratio, NH_4_^+^-N uptake amount, GS activity, and GOGAT activity (*p* < 0.05), whereas they were significantly negatively correlated with the NO_3_^−^-N uptake amount, SP content, and NR activity (*p* < 0.05). Moreover, the NO_3_^−^-N uptake rate exhibited a significant positive correlation with the NO_3_^−^-N uptake amount and NR activity (*p* < 0.05), while showing a significant negative correlation with the GS activity and GOGAT activity (*p* < 0.05) (Figure 4). These results demonstrated that the four emergent plants were influenced by the synergistic effects of environmental adaptation and physiological regulation during N uptake and assimilation. Specifically, a higher NH_4_^+^-N concentration in the water enhanced the activities of GS and GOGAT and increased the NH_4_^+^-N uptake amount, thereby improving the efficiency of NH_4_^+^-N uptake and assimilation. In contrast, in NO_3_^−^-N-enriched water, the plants upregulated the NR activity to enhance their NO_3_^−^-N uptake capacity, and this synergistic mechanism facilitated more efficient utilization of different forms of N in the water [14,50].

The rate of NH_4_^+^-N uptake by plants was significantly and positively correlated with the GS and GOGAT activities. This relationship can be explained using the following perspectives: (1) The GS/GOGAT cycle represents the primary pathway for NH_4_^+^ assimilation in plants, with GS catalyzing the synthesis of glutamine. Subsequently, GOGAT converts glutamine and α-ketoglutarate into glutamic acid, which serves as a critical precursor for the biosynthesis of nitrogenous compounds, including proteins and nucleic acids, and variations in GS activity can serve as a signaling cue for N uptake, thereby regulating N acquisition and utilization in plants [15]. (2) Elevated activities of GS and GOGAT not only enhance NH_4_^+^ uptake and assimilation, but also provide additional N for plant growth, facilitating the synthesis of biological macromolecules such as proteins and nucleic acids, and thus promoting the growth and development of plants [11]. (3) By regulating the expression of key enzymes that are involved in N metabolism, such as GS and GOGAT, plants can effectively enhance their adaptability to environmental stress, thereby promoting growth and productivity [51]. Aquatic plants preferentially absorb NH_4_^+^, which enhances GS and GOGAT activities while suppressing NR and NiR activities, reduce intracellular NH_4_^+^ accumulation, and thus prevent NH_4_^+^ toxicity. This mechanism not only ensures efficient utilization of N sources in water bodies, but also minimizes energy consumption [15].

There are several plausible reasons for the significant positive correlation between the NO_3_^−^-N uptake rate and NR activity, as follows: (1) NR acts as the initial enzyme in the nitrate assimilation process and serves as the rate-limiting step [44]. Higher NR activity increases the plant’s ability to reduce NO_3_^−^ to NH_4_^+^, thereby enhancing the NO_3_^−^ uptake efficiency [52]. (2) Under NO_3_^−^ induction, the expression of NR-related genes is markedly upregulated, resulting in elevated NR activity and consequently improving NO_3_^−^ absorption and assimilation [53]. (3) The uptake of NO_3_^−^ by plants demands a greater energy investment than that required for NH_4_^+^ uptake [22]. When the NR activity is high, it suggests that plants prioritize allocating energy toward NO_3_^−^ uptake to meet their N metabolism needs [17].

In this study, the inorganic N uptake rates of *P. australis*, *T. orientalis*, and *S. validus* were significantly correlated with the SRA and RTD. The main reasons for this correlation may be summarized as follows: (1) Most leaves of *P. australis*, *T. orientalis*, and *S. validus* are positioned above the water layer, precluding direct N absorption. To adapt to their aquatic environment, these plants have evolved well-developed root systems, allowing them to predominantly absorb inorganic N via their roots [54]. (2) Larger SRAs increase the surface area of the root system, thereby enhancing diffusion and mass flow processes and improving the uptake rate of inorganic N [55]. (3) A larger SRA may reflect a greater density of root hairs on the root surface. These root hairs secrete organic acids that desorb adsorbed inorganic N, thus facilitating its absorption by the root system [56]. (4) Plants with lower RTDs tend to exhibit larger SRAs, which enlarge the contact area between their roots and the environment, thereby promoting N diffusion and uptake [57].

In addition, the NH_4_^+^-N uptake rates of *P. australis* and *T. orientalis* were significantly positively correlated with the GDH activity, which may be linked to their ammonium toxicity resistance. This implies that *P. australis* and *T. orientalis* could be better adapted to eutrophic water bodies through the GDH pathway [12]. The inorganic N uptake rates of *P. australis* and *T. orientalis* exhibited a negative correlation with biomass and RTD, suggesting that higher N uptake rates might lead to energy reallocation toward root development, consequently inhibiting aboveground growth [58]. Alternatively, alterations in root structure could potentially affect the N uptake efficiency [57].

### 3.4. The Main Drivers of the Nitrogen Uptake Rates of Emergent Plants

In this study, the N uptake amount of the plants was identified as the most influential factor affecting the overall N uptake rate, with the plants’ N content being particularly significant. Root attributes, specifically the SRA and SRL, were the second most important factors. N assimilation enzymes, including NiR, SP, and NR, had a smaller influence on the N uptake rate. However, photosynthetic parameters exhibited no significant impact on the N uptake rate (Figure 6). These findings confirm our second hypothesis. Furthermore, this study revealed that the following key indicators influence the plants’ N uptake rates: the NO_3_^−^-N uptake amount in stems and leaves, TN content in stems and leaves, NH_4_^+^-N uptake amount in stems and leaves, SRA, NH_4_^+^-N uptake amount in roots, NR activity in stems and leaves, NiR activity in roots, SP content in stems and leaves, SP content in roots, and NR activity in roots (Figure 5).

The N uptake amount exerted the greatest total impact on the N uptake rate, especially the plants’ TN content and NO_3_^−^-N uptake amount in the stem and leaves, which was mainly due to the following dynamics: (1) Plants with higher amounts of N uptake generally have more complete N metabolism pathways and more efficient N transport systems, and they can adjust their N uptake rate more flexibly to adapt to changes in the external N supply when faced with different concentrations of N [59]. (2) The roots of aquatic plants absorb different forms of N from water or sediment and then transport and distribute them to various tissues and organs [60]. When the N contents of the stems and leaves are high, it indicates that the plant has abundant N stores, which can provide sufficient biological macromolecules, such as enzymes and carrier proteins, for related physiological activities, thereby maintaining a high N uptake capacity [61]. (3) Stem and leaf cells have high metabolic activity, can quickly absorb and transport N (their NR activity is especially high), and can reduce NO_3_^−^ to NH_4_^+^, which can then be used by plants [62]. Moreover, the photosynthesis process of stems and leaves produces a large amount of energy and carbon backbone, which provides a material basis and dynamic support for N absorption and assimilation [63]. (4) NO_3_^−^ in water has high solubility and mobility, can spread rapidly in water, and is in full contact with the stems and leaves of aquatic plants, meaning that it is more easily absorbed by the stems and leaves of aquatic plants [64]. Compared with NH_4_^+^, NO_3_^−^ is relatively less toxic to aquatic plants, and they can continuously absorb and accumulate NO_3_^−^ within a certain range [65].

Root traits, especially the SRA and SRL, have a greater impact on N uptake rates, which may be due to the following dynamics: (1) Aquatic plants with larger SRAs and SRLs have better exposure to N in water, thereby increasing their N uptake rates [54,55]. At the same time, plant roots adjust their morphology and physiological functions, such as by increasing their SRA and SRL, to different forms of inorganic N to improve N uptake and utilization [21]. (2) Roots are the main part of N absorption in plants, and root cells are rich in N assimilation enzymes and N transporters. In addition, adjusting the roots’ morphology can provide more substrates for root N assimilation enzymes, and, by adjusting the N assimilation enzyme activity, the absorbed N is converted into transportable and available organic N compounds, while N is transported to the aerial parts through redistribution strategies to meet the growth needs of stems and leaves, which means that it has a greater impact on the N uptake rate [66]. (3) *P. australis*, an emergent plant with well-developed aeration tissue, can promote microbial N conversion through root oxygen secretion, thereby increasing N uptake [32]. In this study, this was also confirmed by a significant positive correlation between the SRA and N uptake rate in the emergent plants (Figure 5).

N assimilation enzymes, particularly NiR, SP, and NR, play a crucial role in influencing N uptake rates. This can primarily be explained by the following factors: (1) The activity of N assimilation enzymes (e.g., NR and GS) serves as a key indicator of the metabolic efficiency of NO_3_^−^ and NH_4_^+^ in plants. Under varying N sources, the activity of N assimilation enzymes in aquatic plants adjusts dynamically to adapt to environmental conditions. Specifically, under high-NO_3_^−^ conditions, the NR activity is significantly increased. In contrast, under high-NH_4_^+^ conditions, the NR activity is markedly inhibited, while the GS expression is upregulated [13,14]. Moreover, soluble proteins, which include a variety of N-assimilating enzymes (e.g., NR and GS), play a critical role in influencing N uptake rates by modulating the activities of the key enzymes that are involved in N metabolism. Soluble proteins also function as a form of storage of N and synergistically interact with photosynthesis [67]. These findings are consistent with the significant positive correlation that we observed between the N assimilation enzyme activity and N uptake rates in emergent plants (Figure 5). (2) The nitrate assimilation process constitutes a critical pathway for plants to achieve sustainable growth and enhance their productivity. In this process, NR and NiR function as pivotal enzymes [10]. NR serves as the initial enzyme in the nitrate assimilation process and acts as the rate-limiting enzyme that catalyzes the reduction of NO_3_^−^ to NO_2_^−^. Subsequently, NiR reduces NO_2_^−^ to NH_4_^+^, thereby enabling further involvement in the synthesis of amino acids and proteins [46]. However, when NH_4_^+^ accumulates at high levels in plants, it may inhibit the activity of NR and NiR by modulating the cellular redox state or signal transduction pathways, thus reducing the demand for NO_3_^−^ reduction and conserving energy [35].

Since the samples in this study were collected at a single time point, they could only reflect the uptake rates of different N forms by emergent plants during a specific period and did not consider the dynamic regulation of enzyme activity throughout the plants’ growth cycle. Consequently, this study has certain limitations. Furthermore, the test water consisted of a ^15^N isotopically labeled NH_4_^+^/NO_3_^−^ proportional gradient solution, which is relatively simplified compared with natural water bodies and neglects the influence of other environmental factors (e.g., light, temperature, pH, and flow rate). In the future, we aim to conduct more comprehensive investigations into the N uptake rates and influencing factors of aquatic plants across different seasons using in situ field experiments.

## 4. Materials and Methods

### 4.1. Experimental Design

The ^15^N isotope labeling experiment was carried out at the Key Laboratory of Ningxia Academy of Agriculture and Forestry Sciences, Ministry of Agriculture and Rural Affairs. Considering the survival rate, ecotype, and purification efficiency, seedlings of four common emergent plants in the Yellow River irrigation area of Ningxia—*P. australis*, *T. orientalis*, *S. validus*, and *L. salicaria*—were transplanted into an agricultural drainage ditch and grown for two months as experimental materials.

Based on previous studies [14,68] and preliminary monitoring conducted by our research group, the NH_4_^+^/NO_3_^−^ ratio was observed to range from 0.41 to 3.67. A N-free Hoagland nutrient solution was used as the base solution, and five NH_4_^+^/NO_3_^−^ gradients with identical total N concentrations were established for conducting ^15^N isotope labeling experiments (Table 2). Each emergent plant species was divided into two groups, as follows: one group was prepared using ^15^NH_4_Cl (99.12 atom%) and NaNO_3_, according to the five NH_4_^+^/NO_3_^−^ gradients, while the other group was prepared using Na^15^NO_3_ (99.21 atom%) and NH_4_Cl. Each treatment was replicated three times, resulting in a total of 30 experimental beakers. Each beaker contained 1 L of test solution with a uniform total N concentration of 15 mg N L^−1^. Plant transplantation was initiated at 10:00 on 2 July 2023, and the experiment continued for 24 h.

At 10:00 on July 3, the net photosynthetic rate of the plant leaves was measured using a plant photosynthetic analyzer (CIRAS-3, PP Systems, Amesbury, MA, USA), with three replicates per treatment and three plants being measured per replicate. After the completion of the ^15^N isotope labeling experiment, the chlorophyll content of the leaves was determined; roots were scanned for the root attribute analysis; and the dry weight, N content, SP content, N assimilation activity, and ^15^N atomic percentages of the stem, leaf, and root samples were analyzed.

### 4.2. Sampling and Measurements

Following the completion of the ^15^N isotope labeling experiment, the plants were carefully rinsed with distilled water and subsequently separated into two components for further analysis: the aboveground biomass (stems and leaves) and the belowground biomass (roots).

The chlorophyll a (CHA) and chlorophyll b (CHB) contents were determined using 95% ethanol extraction. Specifically, 0.5 g of fresh leaf samples was accurately weighed and transferred into a 10 mL centrifuge tube, followed by the addition of 10 mL of 95% ethanol. The mixture was incubated for 48 h at room temperature in the dark. Subsequently, the supernatant was carefully collected, and the absorbance values at wavelengths of 665 nm and 649 nm were measured using a multifunctional microplate reader (Thermo Varioskan LUX, Shanghai, China). Finally, the concentrations of CHA, CHB, and total chlorophyll (CHT) were calculated based on the measured absorbance values.

Determination of Root Functional Traits: Roots were dissected following their branching order, carefully arranged on clear glass without overlapping, scanned with an Epson desktop scanner (Epson Expression 10 000 XL, Epson, Suwa, Japan), and analyzed using the root analysis software WinRHIZO (v. 2020a, Regent Instruments, Quebec, QC, Canada). Subsequently, the AD, total root length, root surface area, root volume, and number of root segments for each plant were determined.

Determination of Enzyme Activity in Plant Stems, Leaves, and Roots: Fresh samples of the plants’ stems, leaves, and roots were carefully cleaned, finely minced, thoroughly homogenized, and accurately weighed at 0.1 g of fresh weight. The samples were subsequently ground into a fine powder under liquid N conditions. The SP content was quantified using a bicinchoninic acid (BCA) protein assay kit. The activities of NR, NiR, GS, GOGAT, and GDH were quantified using a double-antibody one-step sandwich enzyme-linked immunosorbent assay (ELISA).

Determination of N Content in Different Forms of Stems, Leaves, and Roots of the Plants: Fresh samples of the plants’ stems, leaves, and roots were dried at 65 °C for 48 h until a constant weight was achieved and then accurately weighed. Subsequently, the samples were ground into a fine powder using a ball mill (MM2, Fa. Retsch, Haan, Germany) and passed through a 100-mesh sieve, and the ^15^N atomic percentage was analyzed using an isotope-ratio mass-spectrometer–elemental-analyzer (IRMS-EA, Elementar, Manchester, UK).

The TN content of the plant samples was determined using the Kjeldahl N determination method.

### 4.3. Calculations of Various Indicators

The contents of CHA, CHB, and CHT can be calculated using Equations (1)–(3).(1)$$CHA=13.95\times A_{665}-6.88\times A_{649}$$(2)$$CHB=24.96\times A_{649}-7.32\times A_{665}$$(3)CHT=CHA+CHB
where A_665_ and A_649_ represent the absorbance values of the chlorophyll extract at wavelengths of 665 nm and 649 nm, respectively; CHA denotes the concentration of chlorophyll a (mg L^−1^); CHB represents the concentration of chlorophyll b (mg L^−1^); and CHT indicates the total chlorophyll concentration (mg L^−1^).

Equations (4)–(6) for the SRL, SRA, and RTD are expressed as follows:(4)SRL=L/W(5)SRA=S/W(6)RTD=W/V
where SRL denotes the specific root length (m g^−1^); SRA represents the specific root surface area (cm^2^ g^−1^); RTD indicates the root tissue density (g cm^−3^); and L, W, S, and V correspond to the total length (m), dry weight (g), surface area (cm^2^), and volume (cm^3^) of each root segment, respectively.

Equations (7) and (8) can be used to calculate the plants’ N uptake rate and the N uptake amount of each organ of the plants’ roots, stems, and leaves, and are expressed as follows [69]:(7)$$NUR=\frac{({APE}_{R}\times N_{R}\times W_{R}+{APE}_{L}\times N_{L}\times W_{L})\times S_{N}}{W_{R}\times T\times A_{N}}$$(8)$${NUA}_{X}=\frac{{APE}_{X}\times N_{X}\times W_{X}\times S_{N}}{A_{N}}$$
where NUR denotes the rate of N uptake by the plants (μg N g^−1^·h^−1^ DW); ${APE}_{R}$ and APE_L_ represent the ^15^N atom percent excess in the roots and the stems and leaves, respectively; N_R_ and N_L_ (µg N g^−1^) indicate the N contents in the roots and the stems and leaves, respectively; W_R_ and W_L_ (g) refer to the dry weights of the roots and the stems and leaves, respectively; T represents the labeling duration (h); NUA_X_ signifies the N uptake by each organ (root, stem, and leaf) of the plant (μg·N), with the subscript X denoting the specific organ (root, stem, and leaf); S_N_ refers to the N concentration of NH_4_^+^ or NO_3_^−^ in each beaker (mmol L^−1^); and A_N_ indicates the ^15^N concentration (mmol L^−1^) of the added labeling solution.

### 4.4. Statistical Analysis

The effects of N form, plant species, and NH_4_^+^/NO_3_^−^ on plant N uptake rate were analyzed with a generalized linear model (GLM) with gamma distribution and identity link. Model effects were assessed using Type III Wald χ^2^ tests to independently evaluate the significance of each variable and interaction term. The parameters of the GLM were estimated via the maximum likelihood estimation method. The difference was tested using the least squares method (LSD), and the significance level was *p* < 0.05. The differences in N uptake rate and enzyme activities of N assimilation between different treatments were determined to be statistically significant at a level of *p* < 0.05 using the one-way ANOVA and LSD tests. A Pearson correlation analysis was employed to examine the associations among the N uptake rate and the various influencing factors, including N uptake amounts (Appendix A), root traits and photosynthetic parameters (Appendix A). The RDA was employed to examine the relationships between the rate of N uptake and the influencing factors. The PLS-PM showed the effects of the N uptake amount, traits of roots, assimilation enzymes, and photosynthetic parameters on the N uptake rate. All the statistical analyses were carried out using R software (version 3.1.2).

## 5. Conclusions

The form of N and the plant species significantly influenced the N uptake rates of four emergent plant species in the Yellow River irrigation area. In water bodies with varying NH_4_^+^/NO_3_^−^ ratios, the N uptake rates of *P. australis* and *T. orientalis* were significantly higher than those of *S. validus* and *L. salicaria*. The four emergent plant species’ preferences for different forms of N exhibited significant environmental plasticity. At an NH_4_^+^/NO_3_^−^ ratio of 5:5, *P. australis* and *T. orientalis* showed a preference for NO_3_^−^-N absorption, whereas *S. validus* and *L. salicaria* favored NH_4_^+^-N uptake. The N uptake and assimilation processes in the four plants involved a synergistic mechanism of environmental adaptation and physiological regulation, enabling them to absorb different forms of N from water more effectively. Specifically, the NH_4_^+^-N uptake rates of the four emergent plants increased with the enhancement of GS and GOGAT activities, while the NO_3_^−^-N uptake rates increased with the enhancement of NR activity. *P. australis* and *T. orientalis* exhibited enhanced adaptation to eutrophication through the GDH pathway. The amount of N uptake exerted the greatest total impact on the N uptake rate, particularly on the plants’ N content. The second most influential factor was root traits, specifically the SRA and SRL. N assimilation enzymes, especially NiR, SP, and NR, also played a significant role in regulating N uptake.

The divergence in N uptake strategies among species underscores the pivotal role of species selection in ecological restoration. *P. australis* and *T. orientalis* stand out as optimal candidates for remediating N-contaminated waters in the Yellow River Basin due to their high adaptability and efficient N assimilation capacity. The establishment of complementary communities with species such as *S. validus* and *L. salicaria* can further enhance the resilience of plant assemblages to complex N-polluted environments. Moreover, synergistic optimization of plant N assimilation efficiency can be achieved by enhancing enzyme activity through exogenous carbon supplementation, expanding root absorption surfaces via rhizospheric microbial inoculation, and considering environmental factors such as N form and load in water.

Future research should focus on elucidating the seasonal dynamics of N uptake rates and the physiological responses of aquatic plants in natural water bodies. Establishing long-term monitoring networks to track seasonal variations and community succession effects will advance our understanding of the regulatory mechanisms underlying N uptake efficiency. Such efforts will deepen our knowledge of aquatic plant N uptake characteristics and promote their practical application in addressing water N pollution.

## Figures and Tables

**Figure 1 plants-14-01484-f001:**
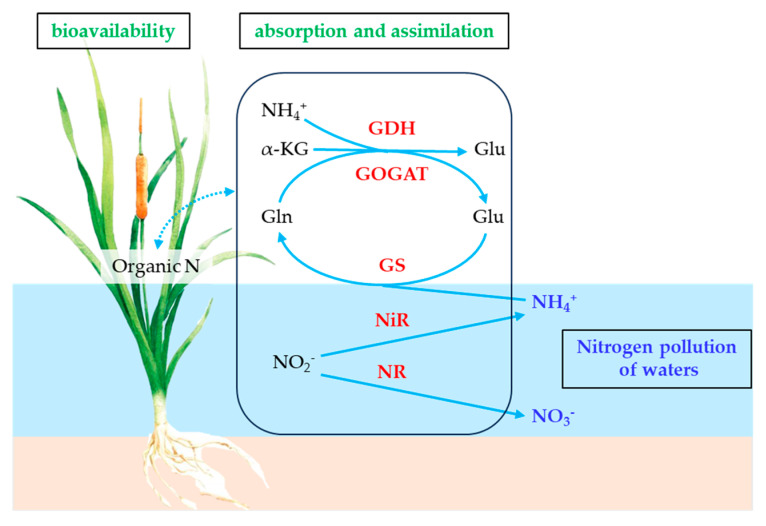
The process of nitrogen uptake and assimilation by plants in aquatic ecosystems. The arrows represent the key nitrogen assimilation processes.

**Figure 2 plants-14-01484-f002:**
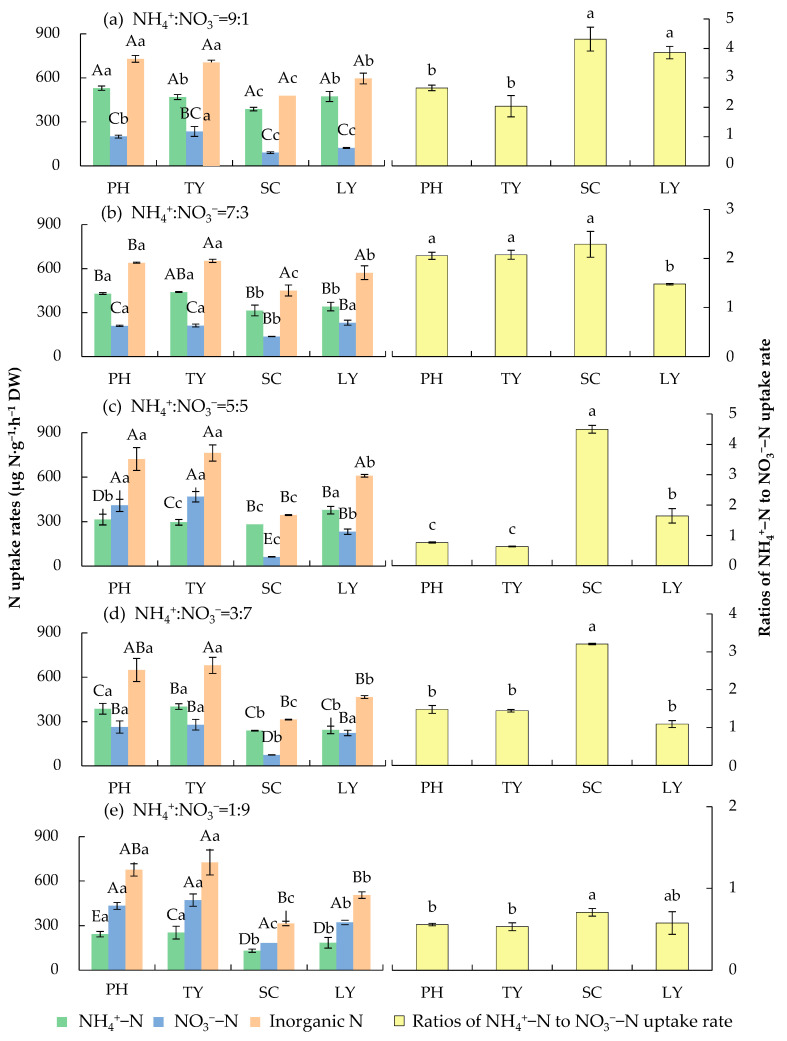
The uptake rates and ratios of NH_4_^+^-N and NO_3_^−^-N of different plants in water with different NH_4_^+^/NO_3_^−^ ratios. PH, *P. australis*; TY, *T. orientalis*; SC, *S. validus*; LY, *L. salicaria*. Different uppercase letters indicate significant differences in the uptake rates of the same nitrogen form by the same plant among varying NH_4_^+^/NO_3_^−^ ratios (9:1, 7:3, 5:5, 3:7, and 1:9) (*p* < 0.05). The different lowercase letters on the left side of the figure indicate significant differences in the uptake rates of the same nitrogen form among different plant species (PH, TY, SC, and LY) at the same NH_4_^+^/NO_3_^−^ ratio (*p* < 0.05). The different lowercase letters on the right side of the figure indicate significant differences in the ratios of NH_4_^+^-N to NO_3_^−^-N uptake rate among different plant species (PH, TY, SC, and LY) at the same NH_4_^+^/NO_3_^−^ ratio (*p* < 0.05).

**Figure 3 plants-14-01484-f003:**
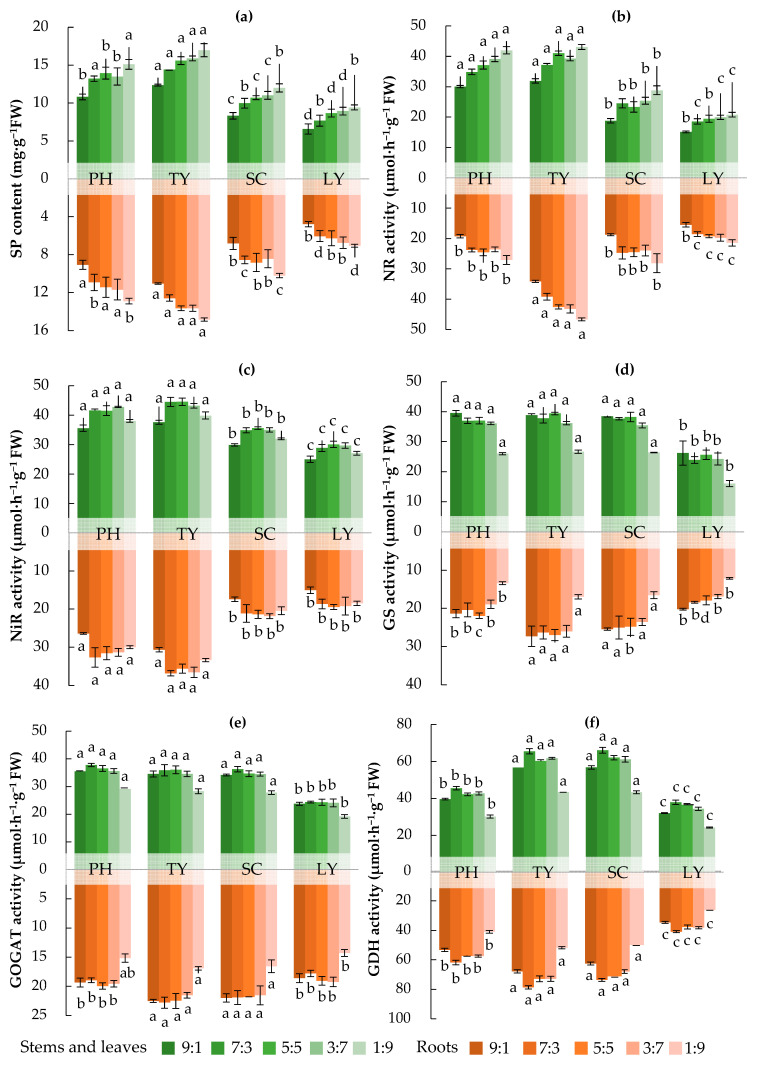
Nitrogen assimilation enzyme activities in stems, leaves, and roots of different plants in water with different NH_4_^+^/NO_3_^−^ ratios. (**a**) SP content, (**b**) NR activity, (**c**) NiR activity, (**d**) GS activity, (**e**) GOGAT activity, (**f**) GDH activity. PH, *P. australis*; TY, *T. orientalis*; SC, *S. validus*; LY, *L. salicaria*. Different lowercase letters indicate significant differences in nitrogen assimilation enzyme activities among the different plants (*p* < 0.05).

**Figure 4 plants-14-01484-f004:**
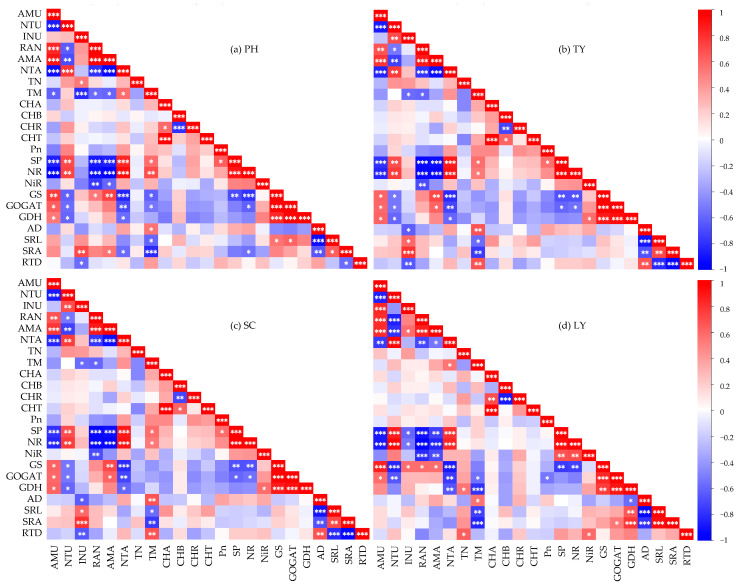
Correlations of factors affecting the nitrogen uptake rate of emergent plants. (**a**) PH, *P. australis*; (**b**) TY, *T. orientalis*; (**c**) SC, *S. validus*; (**d**) LY, *L. salicaria*. AMU, the uptake rate of NH_4_^+^-N; NTU, the uptake rate of NO_3_^−^-N; INU, the uptake rate of inorganic nitrogen; RAN, the NH_4_^+^/NO_3_^−^ ratio; AMA, the uptake amount of NH_4_^+^-N; NTA, the uptake amount of NO_3_^−^-N; TN, the total nitrogen content of a plant; TM, the plant biomass; CHA, the content of chlorophyll a; CHB, the content of chlorophyll b; CHR, the ratio of chlorophyll a to chlorophyll b; CHT, the total chlorophyll content; Pn, the net photosynthetic rate; SP, the content of soluble protein; NR, the activity of nitrate reductase; NiR, the activity of nitrite reductase; GS, the activity of glutamine synthetase; GOGAT, the activity of glutamate synthase; GDH, the activity of glutamate dehydrogenase; AD, the diameter of fine roots; SRL, the specific root length of fine roots; SRA, the specific root surface area of fine roots; RTD, the tissue density of fine roots. Significance codes: *** means *p* < 0.001, ** means *p* < 0.01, * means *p* < 0.05.

**Figure 5 plants-14-01484-f005:**
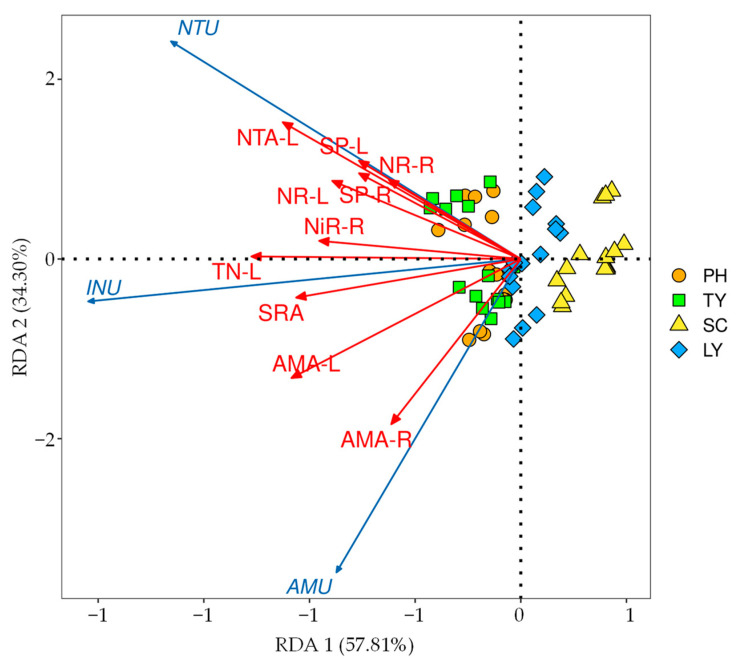
RDA of factors affecting nitrogen uptake rate of emergent plants. AMA-L, uptake amount of NH_4_^+^-N in stems and leaves; AMA-R, uptake amount of NH_4_^+^-N in roots; NTA-L, uptake amount of NO_3_^−^-N in stems and leaves; TN-L, total nitrogen content in stems and leaves; SP-L, content of soluble protein in stems and leaves; SP-R, content of soluble protein in roots; NR-L, activity of nitrate reductase in stems and leaves; NR-R, activity of nitrate reductase in roots; NiR-R, activity of nitrite reductase in roots.

**Figure 6 plants-14-01484-f006:**
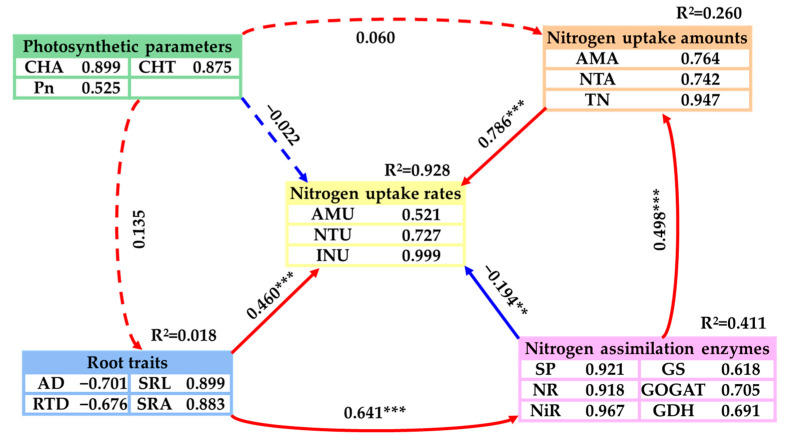
A PLS-PM showing the effects of the nitrogen uptake amounts, root traits, nitrogen assimilation enzymes, and photosynthetic parameters on nitrogen uptake rates. The blue and red lines indicate positive and negative effects, respectively. The solid and dashed arrows indicate significant and insignificant path coefficients, respectively. The numbers adjacent to each arrow denote the path coefficients (significance codes: *** *p* < 0.001, ** *p* < 0.01). The number next to each indicator represents the contribution of each predictor variable to the latent variable. The R^2^ values display the proportion of variance in the nitrogen uptake rate that is explained by each factor.

**Table 1 plants-14-01484-t001:** Generalized linear model (GLM) of the effects of the species, NH_4_^+^/NO_3_^−^ ratio, and nitrogen form on the plants’ nitrogen uptake rates.

Factor	Wald χ^2^	*Df*	*p* Value	Goodness of Fit	Value	*Df*	Value/*Df*
Species	1518.54	3	**<0.001**	Deviance	0.53	80	0.007
NH_4_^+^/NO_3_^−^	91.07	4	**<0.001**	Scaled Deviance	120.09	80	–
N form	609.40	1	**<0.001**	Pearson χ^2^	0.52	80	0.006
Species × NH_4_^+^/NO_3_^−^	119.29	12	**<0.001**	Scaled Pearson χ^2^	118.07	80	–
Species × N form	174.11	3	**<0.001**	Log Likelihood	−511.30	–	–
NH_4_^+^/NO_3_^−^ × N form	1561.83	4	**<0.001**	AIC	1104.61	–	–
Species × NH_4_^+^/NO_3_^−^ × N form	392.85	12	**<0.001**	AICC	1148.76	–	–
(Intercept)	23,275.78	1	**<0.001**	BIC	1218.89	–	–
Omnibus Test (vs. Intercept)	(Likelihood Ratio χ^2^) 477.45	39	**<0.001**	CAIC	1259.89	–	–

Bold *p* values indicate significance. AIC: Akaike Information Criterion; AICC: Corrected Akaike Information Criterion; BIC: Bayesian Information Criterion; CAIC: Consistent AIC. Smaller values indicate better model fit. Wald χ^2^ tests were performed for Type III effects.

**Table 2 plants-14-01484-t002:** Experimental design for ^15^N isotope labeling.

Number	NH_4_^+^/NO_3_^−^	NH_4_Cl (mg L^−1^)	NaNO_3_ (mg L^−1^)
1	9:1	13.5	1.5
2	7:3	10.5	4.5
3	5:5	7.5	7.5
4	3:7	4.5	10.5
5	1:9	1.5	13.5

## Data Availability

The original contributions presented in the study are included in the article/Appendix A.

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
