# Peer review of "Emergent Plants Improve Nitrogen Uptake Rates by Regulating the Activity of Nitrogen Assimilation Enzymes"

_plants, 2025, doi:10.3390/plants14101484_

Round 1
Reviewer 1 Report
Comments and Suggestions for Authors
The study addresses an important and timely topic, specifically“ analyzes the responses of nitrogen uptake rate and physiological characteristics of emergent plants to different forms of nitrogen, clarifies the adaptation strategies of plants to nitrogen forms, reveals the removal mechanism of nitrogen pollution by emergent plants, provides a scientific basis for optimizing the efficient purification of emergent plants and using emergent plants to remediate water nitrogen pollution, and deepens the correlation mechanism,” with clear objectives and a sound methodology. The concept is interesting and technically solid. However, some improvement is needed to improve the overall quality of the manuscript. Below are specific suggestions:
Line 31: Use the full name in the abstract for “GS” and “GOGAT”
Line 36: Use the full form of “GDH”
Figure 2: The letters above (Aa, Bc, Cc etc.) are confused. Please explain them in detail in the figure caption.
I have noticed that authors used some very old references. Please update them with the latest references, not older than 5 years, if possible.
Line 697: Is it Three-way Anova? Did the author check the normality and homogeneity of the data?
Improve the conclusion part by presenting the summary of your study, future research lines, and the application of the study in the field or the environment.
Author Response
Responses to reviewers´ comments
First, we thank the reviewer for the valuable comments and constructive suggestions on our manuscript. We addressed all the comments point-by-point as follows.
Reviewer #1´s comments
The study addresses an important and timely topic, specifically“ analyzes the responses of nitrogen uptake rate and physiological characteristics of emergent plants to different forms of nitrogen, clarifies the adaptation strategies of plants to nitrogen forms, reveals the removal mechanism of nitrogen pollution by emergent plants, provides a scientific basis for optimizing the efficient purification of emergent plants and using emergent plants to remediate water nitrogen pollution, and deepens the correlation mechanism,” with clear objectives and a sound methodology. The concept is interesting and technically solid. However, some improvement is needed to improve the overall quality of the manuscript.
Re. Thanks for the affirmation and valuable suggestion. Based on the comments, we have revised the manuscript to make it better.
Comments 1: Line 31: Use the full name in the abstract for “GS” and “GOGAT”. Line 36: Use the full form of “GDH”.
Response 1:
Thank you for pointing this out. We agree with this comment. Therefore, we have added the following text.
“Glutamine synthetase and Glutamate synthase” (Line 31).
“Glutamate dehydrogenase (GDH)” (Line 37).
Comments 2: Figure 2: The letters above (Aa, Bc, Cc etc.) are confused. Please explain them in detail in the figure caption.
Response 2:
We agree with the reviewer. We have made the following changes.
“Different lowercase letters indicate significant differences in the uptake rates/ratios of the same nitrogen form between different plants (p<0.05). Different uppercase letters indicate significant differences in the uptake rates of the same plant between different NH4+/NO3- ratios (p<0.05).”
was changed to
“Different uppercase letters indicate significant differences in the uptake rates of the same nitrogen form by the same plant among varying NH4+/NO3- ratios (9:1, 7:3, 5:5, 3:7, and 1:9) (p<0.05). The different lowercase letters on the left side of figure indicate significant differences in the uptake rates of the same nitrogen form among different plant species (PH, TY, SC, and LY) at the same NH4+/NO3- ratio (p<0.05). The different lowercase letters on the right side of figure indicate significant differences in the ratios of NH4+-N to NO3--N uptake rate among different plant species (PH, TY, SC, and LY) at the same NH4+/NO3- ratio (p<0.05).” (Lines 211-218)
Comments 3: I have noticed that authors used some very old references. Please update them with the latest references, not older than 5 years, if possible.
Response 3:
Thank you for pointing this out. We agree with this comment. After discussing this issue, we have conducted a thorough search for the latest relevant literature and replaced the majority of the outdated references.
Comments 4: Line 697: Is it Three-way Anova? Did the author check the normality and homogeneity of the data?
Response 4:
We agree with the reviewer. Thank you for the expert's reminder. I apologize for any oversight on my part.
Table 1 presents a three-factor analysis of variance involving three independent variables (Species, NH4+/NO3-, and Nitrogen form). The analysis evaluates their main effects, second-order interactions, and the third-order interaction (e.g., Species×NH4+/NO3-× Nitrogen form).
We conducted supplementary tests for normality (Shapiro-Wilk) (Table R1) and homogeneity of variance (Levene) (Table R2). The results indicate that the data from some groups failed to meet the underlying assumptions. Subsequently, we conducted the Kruskal-Wallis H test. The results revealed that the main effects of Species (H = 30.13, p < 0.001) and Nitrogen form (H = 19.35, p < 0.001) were statistically significant, whereas the main effect of NH4+/NO3- ratio was not significant (Table R3).
Therefore, we applied a generalized linear model (GLM) with gamma distribution and identity link function to analyze the influence of nitrogen uptake rate (Table 1).
The model exhibited an excellent fit (Deviance = 0.53, AIC = 1104.61), with all main effects and interaction terms being highly significant (p < 0.001).
The omnibus test revealed that the current fitted model was significantly different from the intercept-only model (Likelihood Ratio χ2 = 477.45, p < 0.001), confirming the overall validity of the model. The included independent variables (main effects and interaction terms of Species, NH4+/NO3- ratio, and Nitrogen form) were statistically significant in explaining the dependent variable (nitrogen uptake rates).
Based on the above analysis, the following revisions have been made to the paper. Such modifications do not alter the meaning conveyed in the paper.
(1) “Table 1. Multi-way analysis of variance (ANOVA) of the effects of the species, NH4+/NO3- ratio, and nitrogen form on the plants’ nitrogen uptake rates.”
was changed to
“Table 1. Generalized Linear Model (GLM) of the effects of the species, NH4+/NO3- ratio, and nitrogen form on the plants’ nitrogen uptake rates.” (Lines 185-189)
(2) “The effects of N form, plant species and NH4+/NO3- on plant nitrogen uptake rate were analyzed by multivariate ANOVA.”
was changed to
“The effects of N form, plant species and NH4+/NO3- on plant nitrogen uptake rate were analyzed by generalized linear model (GLM) with Gamma distribution and identity link. Model effects were assessed using Type III Wald χ2 tests to independently evaluate the significance of each variable and interaction term. The parameters of the GLM were estimated via the maximum likelihood estimation method.” (Lines 683-687)
Table R1. Shapiro-Wilk normality test. |
||||
Factor |
Group |
Statistic |
Df |
p value |
Species |
PH |
0.926 |
30 |
0.038 |
TY |
0.893 |
30 |
0.006 |
|
SC |
0.910 |
30 |
0.015 |
|
LY |
0.949 |
30 |
0.158 |
|
NH4+/NO3- |
1:9 |
0.923 |
24 |
0.067 |
3:7 |
0.922 |
24 |
0.065 |
|
5:5 |
0.932 |
24 |
0.108 |
|
7:3 |
0.898 |
24 |
0.019 |
|
9:1 |
0.886 |
24 |
0.011 |
|
Nitrogen form |
NH4+-N |
0.981 |
60 |
0.469 |
NO3--N |
0.933 |
60 |
0.003 |
Table R2. Homogeneity of variance (Levene) test. |
|||||
Factor |
Basis of Test |
Statistic |
Df1 |
Df2 |
p value |
Species |
Based on Mean |
0.580 |
3 |
116 |
0.629 |
Based on Median |
0.774 |
3 |
116 |
0.511 |
|
Based on Median with Adjusted Df |
0.774 |
3 |
103.112 |
0.511 |
|
Based on Trimmed Mean |
0.630 |
3 |
116 |
0.597 |
|
NH4+/NO3- |
Based on Mean |
5.328 |
4 |
115 |
0.001 |
Based on Median |
4.825 |
4 |
115 |
0.001 |
|
Based on Median with Adjusted Df |
4.825 |
4 |
109.068 |
0.001 |
|
Based on Trimmed Mean |
5.279 |
4 |
115 |
0.001 |
|
Nitrogen form |
Based on Mean |
0.326 |
1 |
118 |
0.569 |
Based on Median |
0.130 |
1 |
118 |
0.719 |
|
Based on Median with Adjusted Df |
0.130 |
1 |
103.799 |
0.719 |
|
Based on Trimmed Mean |
0.245 |
1 |
118 |
0.621 |
Table R3. Kruskal-Wallis H Test. |
|||
Factor |
Kruskal - Wallis H(K) |
Df |
Asymptotic Significance |
Species |
30.125 |
3 |
<0.001 |
NH4+/NO3- |
2.387 |
3 |
0.496 |
Nitrogen form |
19.3456 |
1 |
<0.001 |
Table 1. Generalized linear model (GLM) of the effects of the species, NH4+/NO3- ratio, and nitrogen form on the plants’ nitrogen uptake rates. |
||||||||
Model Effects |
Wald χ2 |
Df |
p value |
|
Goodness of Fit |
Value |
Df |
Value/Df |
Species |
1518.54 |
3 |
<0.001 |
|
Deviance |
0.53 |
80 |
0.007 |
NH4+/NO3- |
91.07 |
4 |
<0.001 |
|
Scaled Deviance |
120.09 |
80 |
– |
Nitrogen form |
609.40 |
1 |
<0.001 |
|
Pearson χ2 |
0.52 |
80 |
0.006 |
Species × NH4+/NO3- |
119.29 |
12 |
<0.001 |
|
Scaled Pearson χ2 |
118.07 |
80 |
– |
Species × Nitrogen form |
174.11 |
3 |
<0.001 |
|
Log Likelihood |
-511.30 |
– |
– |
NH4+/NO3-×Nitrogen form |
1561.83 |
4 |
<0.001 |
|
AIC |
1104.61 |
– |
– |
Species×NH4+/NO3-×Nitrogen form |
392.85 |
12 |
<0.001 |
|
AICC |
1148.76 |
– |
– |
(Intercept) |
23275.78 |
1 |
<0.001 |
|
BIC |
1218.89 |
– |
– |
Omnibus Test (vs. Intercept) |
(Likelihood Ratio χ2) 477.45 |
39 |
<0.001 |
|
CAIC |
1259.89 |
– |
– |
*Bold p values indicate significance. AIC: Akaike Information Criterion; BIC: Bayesian Information Criterion; CAIC: Consistent AIC. Smaller values indicate better model fit. Wald χ² tests were performed for Type III effects.
Comments 5: Improve the conclusion part by presenting the summary of your study, future research lines, and the application of the study in the field or the environment.
Response 5:
Thank you for pointing this out. We agree with this comment. Therefore, after discussing this issue, the revised conclusion is presented as follows.
“5. Conclusions
The form of nitrogen and plant species significantly influenced the nitrogen uptake rates of four emergent plant species in the Yellow River irrigation area. In water bodies with varying NH4+/NO3- ratios, the nitrogen uptake rates of P. australis and T. orientalis were significantly higher than those of S. validus and L. salicaria. The four emergent plant species’ preference for different forms of nitrogen exhibited significant environmental plasticity. At an NH4+/NO3- ratio of 5:5, P. australis and T. orientalis showed a preference for NO3--N absorption, whereas S. validus and L. salicaria favored NH4+-N uptake. The nitrogen uptake and assimilation processes in the four plants involved a synergistic mechanism of environmental adaptation and physiological regulation, enabling them to absorb different forms of nitrogen from water more effectively. Specifically, the NH4+-N uptake rates of the four emergent plants increased with the enhancement of GS and GOGAT activities, while the NO3--N uptake rates increased with the enhancement of NR activity. P. australis and T. orientalis exhibited enhanced adaptation to eutrophication through the GDH pathway. The amount of nitrogen uptake exerted the greatest total impact on the nitrogen uptake rate, particularly on the plants’ nitrogen content. The second most influential factor was root traits, specifically the SRA and SRL. Nitrogen assimilation enzymes, especially NiR, SP, and NR, also played a significant role in regulating the nitrogen uptake.
The divergence in nitrogen uptake strategies among species underscores the pivotal role of species selection in ecological restoration. P. australis and T. orientalis stand out as optimal candidates for remediating nitrogen-contaminated waters in the Yellow River Basin due to their high adaptability and efficient nitrogen assimilation capacity. The establishment of complementary communities with species such as S. validus and L. salicaria can further enhance the resilience of plant assemblages to complex nitrogen-polluted environments. Moreover, synergistic optimization of plant nitrogen assimilation efficiency can be achieved by enhancing enzyme activity through exogenous carbon supplementation, expanding root absorption surfaces via rhizospheric microbial inoculation, and considering environmental factors such as nitrogen form and load in water.
Future research should focus on elucidating the seasonal dynamics of nitrogen uptake rates and physiological responses of aquatic plants in natural water bodies. Establishing long-term monitoring networks to track seasonal variations and community succession effects will advance our understanding of the regulatory mechanisms underlying nitrogen uptake efficiency. Such efforts will deepen our knowledge of aquatic plant nitrogen uptake characteristics and promote their practical application in addressing water nitrogen pollution.” (Lines 698-731)

Reviewer 2 Report
Comments and Suggestions for Authors
The introduction is adequate and informative. However, it would be more brief and no more than 2.5 pages. Lines 207-226 the uptake ratios for multiple NH₄⁺/NO₃⁻ levels are listed in the text and are not reader-friendly. Instead, a summary table would have been more helpful and would have decreased repetition. Please improve the visibility of the figures (especially fig4). The section of the results supports the conclusions of the research. The research is well structured and overall interesting.
Author Response
Responses to reviewers´ comments
First, we thank the reviewer for the valuable comments and constructive suggestions on our manuscript. We addressed all the comments point-by-point as follows.
Reviewer #2´s comments
The introduction is adequate and informative.
The section of the results supports the conclusions of the research. The research is well structured and overall interesting.
Re. Thanks for the affirmation and valuable suggestion. Based on the comments, we have revised the manuscript to make it better.
Comments 1: The introduction is adequate and informative. However, it would be more brief and no more than 2.5 pages.
Response 1:
Thank you for pointing this out. We agree with this comment. Therefore, after discussing this issue, we have made the following changes.
(1) “NR serves as a rate-limiting enzyme for the assimilation of NO3--N, catalyzing the reduction of NO3- to NO2-, with its activity being significantly affected by the form of nitrogen. Under NO3--N supply conditions, the NR activity is high, whereas it is relatively low under NH4+-N supply conditions. NiR is a key enzyme that reduces NO2- to NH4+, and its activity is positively correlated with NR activity. However, in high-NH4 environments, the activities of both NR and NiR may be inhibited [12].”
was changed to
“NR serves as a rate-limiting enzyme for the assimilation of NO3--N, catalyzing the reduction of NO3- to NO2-. NiR is a key enzyme that reduces NO2- to NH4+, and its activity is positively correlated with NR activity [13]. Under NO3--N supply conditions, the NR activity is relatively high. However, in high NH4+ environments, the activities of both NR and NiR may be inhibited [14].” (Lines 91-95)
(2) This sentence has been deleted.
“The GDH activity in aquatic plants is generally higher when NH4+-N is the primary nitrogen source [11].”
(3) “Different forms of nitrogen (NH4+-N, NO3--N, etc.) in water may have different migration and transformation rules and bioavailability, which may lead to different absorption and utilization efficiencies of different forms of nitrogen by aquatic plants [17]. Plant roots adopt different nitrogen uptake and assimilation strategies to sustain normal growth and development in response to changes in the availability and morphology of nitrogen in water [18]. These pathways and strategies are influenced by a combination of factors such as the plant species, nitrogen morphology in the environment, and ammonium-to-nitrate ratio (NH4+/NO3-) [19,20].”
was changed to
“The migration, transformation, and bioavailability of different nitrogen forms (NH4+-N, NO3--N, etc.) in aquatic environments vary, potentially driving aquatic plant roots to adopt distinct strategies for nitrogen uptake and assimilation to sustain normal growth and development [17]. These pathways and strategies are influenced by a combination of factors such as the plant species, nitrogen morphology in the environment, and ammonium-to-nitrate ratio (NH4+/NO3-) [18].” (Lines 103-108)
(4) This sentence has been deleted.
“NH4+-N treatment significantly enhanced rice’s biomass and nitrogen accumulation, demonstrating greater efficacy compared with NO3--N treatment in promoting the growth of rice. Furthermore, mixed nitrogen (NH4+-N and NO3--N) exhibited a superior performance in increasing plants’ biomass and nitrogen use efficiency compared with single nitrogen forms [22].”
(5) This sentence has been deleted.
“Significant differences were observed between the nitrogen uptake preferences of different plant species, primarily characterized by a preference for either NH4+-N or NO3--N. In temperate desert ecosystems, ephemeral plants such as Centaurea pulchella and Lactuca undulata predominantly favor NO3--N absorption, whereas annual plants such as Ceratocarpus arenarius and Suaeda glauca exhibit a stronger affinity for NH4+-N absorption [25]. While plants can efficiently utilize the available forms of nitrogen in their growth medium, their preferences for specific nitrogen forms and rates of uptake may vary depending on the geographic location, soil properties, climatic conditions, and other environmental factors [26]. Soil NH4+-N was the preferred N form by plants in (sub)tropical regions, whereas the preference for NO3--N was significantly higher in high-latitude than low-latitude regions [27]. In a semi-arid grassland in Inner Mongolia, three species of plants (Leymus chinensis, Stipa grandis, and Cleistogenes squarrosa) altered their N uptake preferences in response to different grazing intensities in the early and vigorous growth stages [28].”
Comments 2: Lines 207-226 the uptake ratios for multiple NH₄⁺/NO₃⁻ levels are listed in the text and are not reader-friendly. Instead, a summary table would have been more helpful and would have decreased repetition.
Response 2:
We agree with the reviewer. After discussing this issue, the text in the paper concerning the uptake ratios across multiple NH₄⁺/NO₃⁻ levels has been restructured. Given that the relevant data are already illustrated in Figure 2, it is no longer necessary to include a table. We have made the following modifications.
“When NH4+-N was dominant (NH4+/NO3- ratios of 9:1 and 7:3), P. australis, T. orientalis, S. validus, and L. salicaria exhibited a preference for absorbing NH4+-N. ……When the NH4+/NO3- ratio was 5:5, the four plants showed varying preferences for different nitrogen forms. P. australis and T. orientalis preferred NO3--N absorption, while S. validus and L. salicaria favored NH4+-N absorption, with their respective uptake rate ratios being 0.77, 0.63, 4.50, and 1.64.”
was changed to
“When NH4+-N was dominant (NH4+/NO3- ratios of 9:1 and 7:3), P. australis, T. orientalis, S. validus, and L. salicaria exhibited a preference for absorbing NH4+-N, and the ratios of NH4+-N to NO3--N uptake rates were all greater than 1. When the NH4+/NO3- ratio was 1:9, the four emergent plants preferred to absorb NO3--N, and the ratios of NH4+-N to NO3--N uptake rates were all less than 1. This indicates that the inorganic nitrogen uptake rates of these four species exhibit a certain degree of environmental plasticity. When the NH4+/NO3- ratio was 5:5, the four plants showed varying preferences for different nitrogen forms. P. australis and T. orientalis preferred NO3--N absorption, while S. validus and L. salicaria favored NH4+-N absorption. “(Lines 198-206)
Comments 3: Please improve the visibility of the figures (especially fig4).
Response 3:
Thank you for pointing this out. We agree with this comment. Therefore, the visibility of all the figures has been improved.

Reviewer 3 Report
Comments and Suggestions for Authors
Dear Authors,
The manuscript submitted for review deals with the very important topic of treating water bodies from excess nitrogen using selected emergent plants. The proposed topic refers very well with the content of the whole manuscript. Also worth appreciating is the reliable reporting of how the data were obtained and how they were processed with discussion. The text reads well, i.e. it is logically arranged, the purposefulness of the tasks undertaken is shown and the results are quite precisely analysed. I think a little work could be done on the overall language style - but this is a minor issue, as a well thought-out topic allows one to properly understand the authors' intentions and interpret the results on their own. Minor comments include that already in the abstract there should be an explanation of the acronyms used (e.g.: GS-Glutamine Synthetase). Figure 4 is illegible, i.e. both blurred and needs more description so that it is clear what the symbols for the processes mean, why black and white dots are used once and whether their number really means something to the reader. In general, only part of the description for Figure 4 needs to be expanded or edited to better show the authors' intentions. In the conclusion, the authors show an improvement in the nitrogen uptake rate for selected plants using numbers like, 114.69% or 130.46%. Firstly, such a high accuracy in giving a result is already a kind of misrepresentation of reality (apart from the fact that the plants were in a controlled environment), because a huge amount of measurement data for each individual nitrogen form concentration gradient is needed for such accurate results. Secondly, from a mathematical point of view, the representation of a phenomenon by means of a percentage scale is very clear to everyone, but exceeding the number of 100% is already treated as an error in mathematics. I understand that we encounter this kind of reporting everywhere, but since this is a scientific text, it is more correct to report the change by saying its multiplication with respect to the base data.
Sincerely Yours,
Reviewer
Author Response
Responses to reviewers´ comments
First, we thank the reviewer for the valuable comments and constructive suggestions on our manuscript. We addressed all the comments point-by-point as follows.
Reviewer #3´s comments
The manuscript submitted for review deals with the very important topic of treating water bodies from excess nitrogen using selected emergent plants. The proposed topic refers very well with the content of the whole manuscript. Also worth appreciating is the reliable reporting of how the data were obtained and how they were processed with discussion. The text reads well, i.e. it is logically arranged, the purposefulness of the tasks undertaken is shown and the results are quite precisely analysed. I think a little work could be done on the overall language style - but this is a minor issue, as a well thought-out topic allows one to properly understand the authors' intentions and interpret the results on their own.
Re. Thanks for the affirmation and valuable suggestion. Based on the comments, we have revised the manuscript to make it better.
Comments 1: Minor comments include that already in the abstract there should be an explanation of the acronyms used (e.g.: GS-Glutamine Synthetase).
Response 1:
Thank you for pointing this out. We agree with this comment. Therefore, we have added the following text.
“Glutamine synthetase and Glutamate synthase” (Line 31).
“Glutamate dehydrogenase (GDH)” (Line 37).
Comments 2: Figure 4 is illegible, i.e. both blurred and needs more description so that it is clear what the symbols for the processes mean, why black and white dots are used once and whether their number really means something to the reader. In general, only part of the description for Figure 4 needs to be expanded or edited to better show the authors' intentions.
Response 2:
We agree with the reviewer. Thank you for the expert's reminder. I apologize for any oversight on my part. The visibility of Figure 4 has been improved. All the white and black * in Figure 4 are uniformly converted to white *. The following text was added to the annotation.
“significance codes: *** means p < 0.001, ** means p < 0.01, * means p < 0.05.” (Lines 308-309)
Comments 3: In the conclusion, the authors show an improvement in the nitrogen uptake rate for selected plants using numbers like, 114.69% or 130.46%. Firstly, such a high accuracy in giving a result is already a kind of misrepresentation of reality (apart from the fact that the plants were in a controlled environment), because a huge amount of measurement data for each individual nitrogen form concentration gradient is needed for such accurate results. Secondly, from a mathematical point of view, the representation of a phenomenon by means of a percentage scale is very clear to everyone, but exceeding the number of 100% is already treated as an error in mathematics. I understand that we encounter this kind of reporting everywhere, but since this is a scientific text, it is more correct to report the change by saying its multiplication with respect to the base data.
Response 3:
Thank you for pointing this out. We agree with this comment.
The ranges of 11.83%–114.69% and 14.07%–130.46% in the paper represent the relative increase rate of inorganic nitrogen uptake rates calculated using the formula below (Table R1, R2).
Following our discussion, we determined that the wide range of relative increase rates rendered it inappropriate to express changes in the baseline data as multiples. For example, “The uptake rates of inorganic nitrogen of P. australis and T. orientalis were significantly higher than those of the other two plant species (p < 0.05), with an increase ranging from 0.1 to 1 times.” The sentence is not suitable in this context.
Additionally, presenting the increase in nitrogen uptake rates of the selected plants as percentages was also deemed unsuitable. Consequently, this section has been removed. Such modifications do not alter the meaning conveyed in the paper.
“In water bodies with varying NH4+/NO3- ratios, the nitrogen uptake rates of P. australis and T. orientalis were significantly higher than those of S. validus and L. salicaria”. (Lines 870-883)
Where denotes the relative increase rate (%); represents the higher nitrogen uptake rate of plant A; indicates the lower nitrogen absorption rate of plant B.
Table R1. Average uptake rates of inorganic nitrogen by different plants in water with different NH4+/NO3- ratios (μg N·g-1·h-1DW). |
|||||
Plants |
9:1 |
7:3 |
5:5 |
3:7 |
1:9 |
PH |
729.20 |
639.65 |
722.64 |
649.21 |
675.42 |
TY |
703.61 |
652.48 |
763.71 |
680.48 |
725.02 |
SC |
477.36 |
451.00 |
344.17 |
313.18 |
314.60 |
LY |
595.72 |
571.99 |
609.92 |
465.56 |
506.03 |
PH, P. australis; TY, T. orientalis; SC, S. validus; LY, L. salicaria. |
Table R2. Average relative increase rate of inorganic nitrogen uptake rates for P. australis and T. orientalis (%). |
|||||
Relative increase rate |
9:1 |
7:3 |
5:5 |
3:7 |
1:9 |
The increase rate of PH compared with SC |
52.76 |
41.83 |
109.96 |
107.29 |
114.69 |
The increase rate of PH compared with LY |
22.41 |
11.83 |
18.48 |
39.44 |
33.47 |
The increase rate of TY compared with SC |
47.40 |
44.67 |
121.90 |
117.28 |
130.46 |
The increase rate of TY compared with LY |
18.11 |
14.07 |
25.21 |
46.16 |
43.28 |
